# Klotho deficiency intensifies hypoxia-induced expression of IFN-α/β through upregulation of RIG-I in kidneys

**Asako Urabe[1], Shigehiro Doi[1]\*, Ayumu Nakashima[1,2], Takeshi Ike[1], Kenichi Morii[1], Kensuke Sasaki[1], Toshiki Doi[1], Koji Arihiro[3], Takao Masaki[1]\***

**1** Department of Nephrology, Hiroshima University Hospital, Hiroshima, Japan, **2** Department of Stem Cell Biology and Medicine, Graduate School of Biomedical & Health Sciences, Hiroshima University, Hiroshima, Japan, **3** Department of Anatomical Pathology, Hiroshima University Hospital, Hiroshima, Japan

\* sdoi@hiroshima-u.ac.jp (SD); masakit@hiroshima-u.ac.jp (TM)

**Data Availability Statement:** All relevant data are within the manuscript and its Supporting Information files.

## Abstract

Hypoxia is a common pathway to the progression of end-stage kidney disease. Retinoic acid-inducible gene I (RIG-I) encodes an RNA helicase that recognizes viruses including SARS-CoV2, which is responsible for the production of interferon (IFN)-α/β to prevent the spread of viral infection. Recently, RIG-I activation was found under hypoxic conditions, and klotho deficiency was shown to intensify the activation of RIG-I in mouse brains. However, the roles of these functions in renal inflammation remain elusive. Here, for *in vitro* study, the expression of RIG-I and IFN-α/β was examined in normal rat kidney (NRK)-52E cells incubated under hypoxic conditions (1% O$_2$). Next, siRNA targeting RIG-I or scramble siRNA was transfected into NRK52E cells to examine the expression of RIG-I and IFN-α/β under hypoxic conditions. We also investigated the expression levels of RIG-I and IFN-α/β in 33 human kidney biopsy samples diagnosed with IgA nephropathy. For *in vivo* study, we induced renal hypoxia by clamping the renal artery for 10 min in wild-type mice (WT mice) and Klotho-knockout mice (Kl$^{-/-}$ mice). Incubation under hypoxic conditions increased the expression of RIG-I and IFN-α/β in NRK52E cells. Their upregulation was inhibited in NRK52E cells transfected with siRNA targeting RIG-I. In patients with IgA nephropathy, immunohistochemical staining of renal biopsy samples revealed that the expression of RIG-I was correlated with that of IFN-α/β (r = 0.57, *P*<0.001, and r = 0.81, *P*<0.001, respectively). The expression levels of RIG-I and IFN-α/β were upregulated in kidneys of hypoxic WT mice and further upregulation was observed in hypoxic Kl$^{-/-}$ mice. These findings suggest that hypoxia induces the expression of IFN-α/β through the upregulation of RIG-I, and that klotho deficiency intensifies this hypoxia-induced expression in kidneys.

## Introduction

Chronic kidney disease (CKD) affected 697.5 million people globally in 2017 [1], so it is well recognized as a major health concern. To retard the progression of CKD, inhibitors of the renin–angiotensin–aldosterone system (RAAS) pathway are used in a clinical setting [2,3].

**Funding:** The author(s) received no specific funding for this work.

**Competing interests:** The authors have declared that no competing interests exist.

However, their beneficial effects are limited and many patients eventually require renal replacement therapy [4]. Pathologically, chronic inflammation and interstitial fibrosis are the common features of CKD, regardless of the primary disease [5]. Although transforming growth factor (TGF)-β1 plays a pivotal role in the development of interstitial fibrosis [6–8], previous studies demonstrated that the infiltration of inflammatory cells is responsible for the production of TGF-β1 [9]. These findings suggest that inflammation is the upstream event of interstitial fibrosis, and that inflammation is a candidate therapeutic target for CKD.

Hypoxia is a condition in which insufficient oxygen makes it into tissues. As a cellular response to hypoxia, hypoxia-inducible factors (HIF) play important roles in preventing tissue damage [10–12]. However, severe hypoxia leads to acute vascular diseases, such as stroke [13], angina pectoris [14], and peripheral artery disease [15]. Recently, it has been reported that chronic hypoxia, such as sleep apnea, also contributes to the development of various diseases [16]. Regarding the kidney, hypoxia reportedly intensifies with the progression of CKD stage and it is currently considered as a common pathophysiology leading to end-stage kidney disease [17]. As mentioned above, inflammation is considered as an upstream event, so the mechanism by which hypoxia induces inflammation should be clarified at the molecular level.

Inflammation is characterized by pain, heat, redness, swelling, and loss of function [18]. However, it basically functions to eliminate the initial cause of cell injury and to clear out necrotic cells and damaged tissue. In any case, inflammation is not a specific response, and innate immunity is considered to mainly participate in the initial process of inflammation through the production of interferon (IFN)-α/β [19]. Among the factors involved in innate immunity, retinoic acid-inducible gene I (RIG-I) is responsible for immune responses to viral infections [20]. Notably, recent studies have demonstrated that hypoxia upregulates RIG-I expression in brain and liver, resulting in the production of IFN-α/β [21,22]. However, the role of hypoxia in the RIG-I-mediated expression of IFN-α/β in kidney remains elusive.

Klotho was first reported as an anti-aging protein [23]. Indeed, klotho-deficient mice exhibit a shortened lifespan accompanied by phenotypes resembling human aging. In contrast, klotho transgenic mice show an extended lifespan [24]. In terms of renoprotective effects, klotho deficiency has been found to intensify renal damage in experimental models of renal diseases, whereas klotho overexpression or the administration of klotho protein ameliorates it [25]. Klotho protein exists in three forms: membrane, secreted, and intracellular klotho. Among these, intracellular klotho is reported to suppress RIG-I expression along with inflammation [26], raising the possibility that klotho deficiency exacerbates hypoxia-induced upregulation of RIG-I and IFN-α/β in kidneys. These findings led us to hypothesize that the expression of RIG-I and IFN-α/β increased under hypoxic conditions in a cell line of renal tubular cells, and that their hypoxia-induced upregulation was intensified in klotho-knockout mice (Kl$^{-/-}$ mice).

In this study, we show that hypoxia upregulates RIG-I as well as IFN-α/β in normal rat kidney cells (NRK-52E) and mouse kidneys. We also show that RIG-I is responsible for the hypoxia-induced upregulation of IFN-α/β. In actual human kidney samples from patients diagnosed with IgA nephropathy, RIG-I expression was found to be positively correlated with IFN-α/β. Lastly, hypoxia-induced upregulation of RIG-I and IFN-α/β was intensified in Kl$^{-/-}$ mice. These findings suggest that klotho deficiency intensifies hypoxia-induced expression of IFN-α/β through the upregulation of RIG-I in kidneys.

## Materials and methods

### Animals

Male C57BL/6J wild-type (WT) mice (aged 8 weeks and weighing 20–25 g) were obtained from Charles River Laboratories Japan (Yokohama, Japan). Male Kl$^{-/-}$ mice (aged 6 weeks and

weighing approximately 15 g) were purchased from CLEA Japan, Inc. The mice were housed in the Institute of Laboratory Animal Science of Hiroshima University (Hiroshima, Japan), as previously described [27]. All experiments were approved by the Institutional Animal Care and Use Committee of Hiroshima University (permit numbers: A19-158 and 2019–156) and were performed in accordance with the National Institutes of Health (NIH) Guidelines on the Use of Laboratory Animals. The WT and $Kl^{-/-}$ mice were divided into a control group or a hypoxia group (n = 5 in each group). Hypoxic conditions were induced by clamping of the left renal artery for 10 min under general anesthesia (0.3 mg/kg medetomidine, 4 mg/kg midazolam, and 5 mg/kg butorphanol) [28]. Mice in the control group underwent a sham operation that was the same procedure as that in mice in the hypoxia group, except for the lack of artery ligation. After 10 min of hypoxia, the mice were euthanized by cardiac puncture and their kidneys were harvested. Hypoxia of the kidney was confirmed by western blotting for erythropoietin expression (S1 Fig).

## Cell culture

The normal rat kidney (NRK)-52E cells were purchased from the American Type Culture Collection (Manassas, VA, USA). NRK-52E cells were cultured in RPMI-1640 medium containing 10% fetal bovine serum (FBS) (Nichirei Bio Science, Tokyo, Japan) and penicillin/streptomycin (Nacalai Tesque, Kyoto, Japan). These cells were seeded into 10 cm culture dishes. After growing to subconfluence, the cells were incubated in a 1% $O_2$ Modular Incubator Chamber (MIC 101; Billups-Rothenberg, San Diego, CA, USA) for 30, 60, 90, and 120 min. Whole-cell lysates were prepared and subjected to western blot analysis. Hypoxia of NRK-52E cells was confirmed by western blotting for HIF-1α expression (S2 Fig).

## Transfection of RIG-I siRNA

NRK52E cells were transfected with 20 nM siRNA targeting RIG-I (RSS311418; Thermo Fisher Scientific, Waltham, MA, USA) or negative control siRNA (4390843; Applied Biosystems, Waltham, MA, USA) using Lipofectamine 2000 Transfection Reagent (Thermo Fisher Scientific), in accordance with the manufacturer's instructions. After 43–45 h, some of these cells were incubated under hypoxic conditions (1% $O_2$) for 30 min to evaluate RIG-I. The other cells were changed to fresh medium. After 24 h, these cells were incubated under hypoxic conditions (1% $O_2$) for 60 and 120 min to confirm the suppression of IFN-α/β. Whole-cell lysates were prepared and subjected to western blot analysis.

## Western blot analysis

Sample collection and western blotting were performed in accordance with previously described methods [29]. Rabbit monoclonal anti-RIG-I antibody (#3743S; Cell Signaling Technology, Danvers, MA, USA), rabbit polyclonal anti-IFN-α 11 antibody (bs-7023R; Bioss Antibodies, Woburn, MA, USA), rabbit polyclonal anti-IFN-β antibody (GTX37658; GeneTex, Irvine, CA, USA), rabbit monoclonal anti-RIG-I antibody (#700366; Invitrogen, Carlsbad, CA, USA), rabbit polyclonal anti-IFN-α2 antibody (ab193055; Abcam, Cambridge, UK), rabbit polyclonal anti-IFN-β antibody (PA5-20390; Invitrogen), rat monoclonal anti-human klotho antibody (KO603; TransGenic, Fukuoka, Japan), mouse monoclonal anti-β-actin antibody (A5316; Sigma-Aldrich, St. Louis, MO, USA), and mouse monoclonal anti-α-tubulin antibody (T9026; Sigma-Aldrich) were used as primary antibodies. Horseradish peroxidase-conjugated goat anti-rabbit immunoglobulin G (Dako, Glostrup, Denmark) and goat anti-mouse immunoglobulin G (Dako) were used as secondary antibodies. SuperSignal West Dura and the Pico system (Thermo Fisher Scientific) were used to detect signals. The intensity of each band was

measured by ImageJ software (version 1.47v; National Institutes of Health, Bethesda, MD, USA) and normalized to the level of either β-actin or α-tubulin.

### Immunohistochemical analysis of mouse kidney tissue

Immunohistochemical staining was performed in accordance with previously described methods [27]. As primary antibodies, the following products were applied: rabbit monoclonal anti-RIG-I antibody (#700366; Invitrogen), rabbit polyclonal anti-IFN-α 2 antibody (ab193055; Abcam), rabbit polyclonal anti-IFN-β antibody (PA5-20390; Invitrogen), and rat monoclonal anti-human klotho antibody (KO603; TransGenic). RIG-I-, IFN-α/β-, and klotho-positive areas were quantified as the average of 20 randomly selected fields with ImageJ software.

### Clinical sample collection and ethics statement

Kidney specimens were obtained by renal biopsy at Hiroshima University Hospital between August 2014 and November 2016 from 33 patients who were diagnosed with IgA nephropathy. This study adhered to the principles embodied in the Declaration of Helsinki and was approved by the Ethics Committee of Hiroshima University (E-1718). Informed consent was obtained in the form of opt-out on a website (https://jinzounaika.hiroshima-u.ac.jp/research/opt_out.html).

### Immunohistochemical analysis of human kidney tissue

Immunostaining was carried out in accordance with previously described methods [27]. The following primary antibodies were used: rabbit monoclonal anti-RIG-I antibody (#700366; Invitrogen), mouse monoclonal anti-IFN-α antibody (sc-373757; Santa Cruz Biotechnology, Dallas, TX, USA), and rabbit polyclonal anti-IFN-β antibody (PA5-20390; Invitrogen). RIG-I- and IFN-α/β-positive areas were quantified as the average of five randomly selected fields with ImageJ software.

### Statistical analysis

Results are presented as mean ± standard deviation (SD). Correlations were calculated using univariate regression analysis. For multiple comparisons, we used one-way analysis of variance (ANOVA) followed by Student's *t*-test with Bonferroni correction. Differences between two groups were analyzed by Student's *t*-test. $P < 0.05$ was considered as statistically significant.

## Results

### Hypoxia upregulates RIG-I and IFN-α/β in a rat cell line of renal epithelial cells

To identify the effect of hypoxia on the expression of RIG-I and IFN-α/β *in vitro*, we performed western blot analysis for these proteins in NRK-52E cells cultured under hypoxic conditions. The protein level of RIG-I peaked at 30 min and then gradually decreased in NRK-52E cells under hypoxic stimulation (Fig 1A). IFN-α/β levels increased and peaked at 120 and 60 min, respectively, in NRK-52E cells with hypoxic stimulation (Fig 1B and 1C).

### Knockdown of RIG-I reduces the expression of IFN-α/β in NRK52E cells with hypoxia

To assess whether RIG-I participates in the expression of IFN-α/β under hypoxic conditions, we investigated their expression in NRK-52E cells transfected with *RIG-I* siRNA or negative

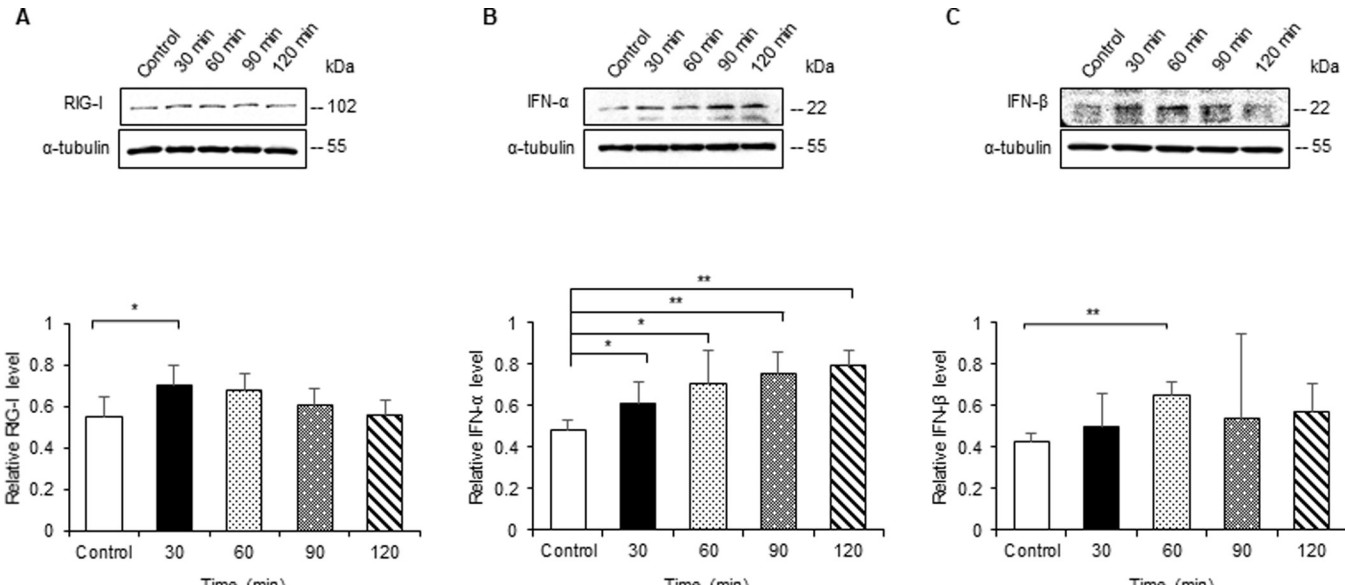

**Fig 1. Hypoxia enhances expression of RIG-I and IFN-α/β in a rat cell line of epithelial cells of renal tubules.** NRK-52E cells were incubated under hypoxic conditions (1.0% $O_2$) for 30, 60, 90, and 120 min. Cell lysates were subjected to western blot analysis using antibodies against RIG-I and IFN-α/β. Typical western blot analysis demonstrated the expression levels of RIG-I (A) and IFN-α/β (B and C). Graphs show the expression levels quantified by densitometry and normalized to α-tubulin (n = 5 in each group). Values are expressed as the mean ± SD. Statistical analysis was performed using ANOVA followed by Tukey's post hoc test. $^*P < 0.05$, $^{**}P < 0.01$.

control siRNA. First, we confirmed the knockdown effect of *RIG-I* siRNA in NRK-52E cells. Western blotting for RIG-I revealed that *RIG-I* siRNA downregulated RIG-I expression in NRK-52E cells either with or without hypoxic stimulation (Fig 2A). We next investigated the role of hypoxia-induced upregulation of RIG-I in the expression of IFN-α/β in NRK-52E cells using *RIG-I* siRNA or negative control. *RIG-I* siRNA suppressed the expression of IFN-α/β in NRK-52E cells (Fig 2B and 2C).

## RIG-I correlates with expression levels of IFN-α/β in human kidney specimens of IgA nephropathy

We examined whether the expression level of RIG-I is associated with IFN-α/β in renal biopsy samples obtained from patients with IgA nephropathy (n = 33). The detailed clinical characteristics of these patients are shown in S1 Table. Immunohistochemical staining for RIG-I and IFN-α/β revealed their expression in glomeruli and renal tubules (Fig 3A), and showed that RIG-I is positively correlated with IFN-α/β (r = 0.57, $P<0.001$, and r = 0.81, $P<0.001$, respectively) (Fig 3B).

## Hypoxia does not change expression level of klotho in WT and $Kl^{-/-}$ mice

To determine the effect of hypoxic stimulation on klotho expression *in vivo*, we investigated its expression in WT and $Kl^{-/-}$ mice with or without 10 min of hypoxic stimulation. Western blotting and immunohistochemical staining revealed that klotho expression did not differ between WT mice with and without hypoxia, and that klotho expression was not observed in $Kl^{-/-}$ mice regardless of the presence or absence of hypoxic stimulation (Fig 4A and 4B). In the WT mice, klotho protein was stained in the cytoplasm of tubular cells (Fig 4B).

**RIG-I expression is intensified under hypoxic conditions in $Kl^{-/-}$ mice.** Intracellular klotho reportedly confers the ability to suppress RIG-I-mediated inflammation, so we

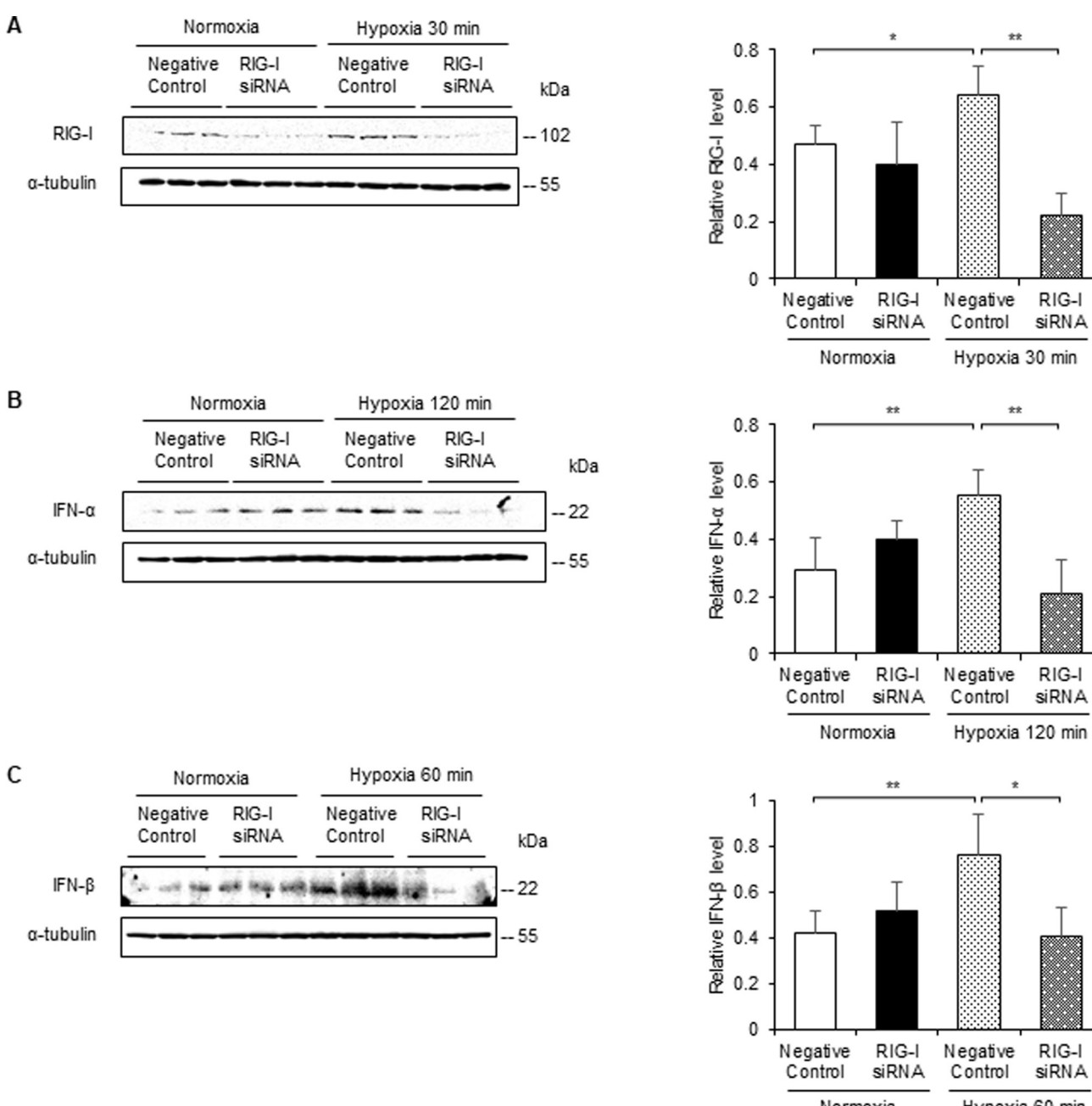

**Fig 2. RIG-I siRNA transfection attenuates expression of IFN-α/β in NRK-52E cells cultured under hypoxic conditions.** NRK-52E cells were transfected with RIG-I siRNA or negative control siRNA. Cell lysates were subjected to western blot analysis using antibodies against RIG-I and IFN-α/β. Typical western blot analysis demonstrated the expression levels of RIG-I (A) and IFN-α/β (B and C). Graphs show the expression levels quantified by densitometry and normalized to α-tubulin (n = 5 in each group). Values are expressed as the mean ± SD. Statistical analysis was performed using ANOVA followed by Tukey's post hoc test. $^{*}P < 0.05$, $^{**}P < 0.01$.

examined the expression level and localization of RIG-I in hypoxic kidneys of WT and Kl$^{-/-}$ mice. Western blotting revealed that RIG-I expression increased in WT mice with hypoxic stimulation, and that it was intensified in Kl$^{-/-}$ mice (Fig 5A). Immunohistochemistry showed that the RIG-I-positive area increased in Kl$^{-/-}$ mice, which was similar to the results obtained from western blotting (Fig 5B).

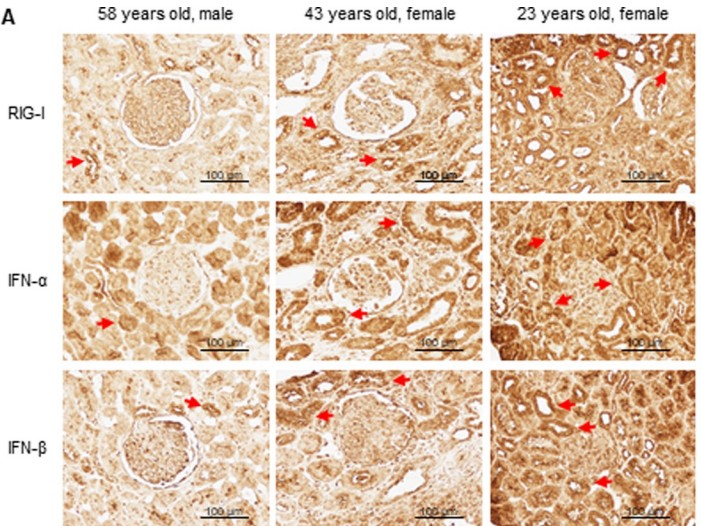

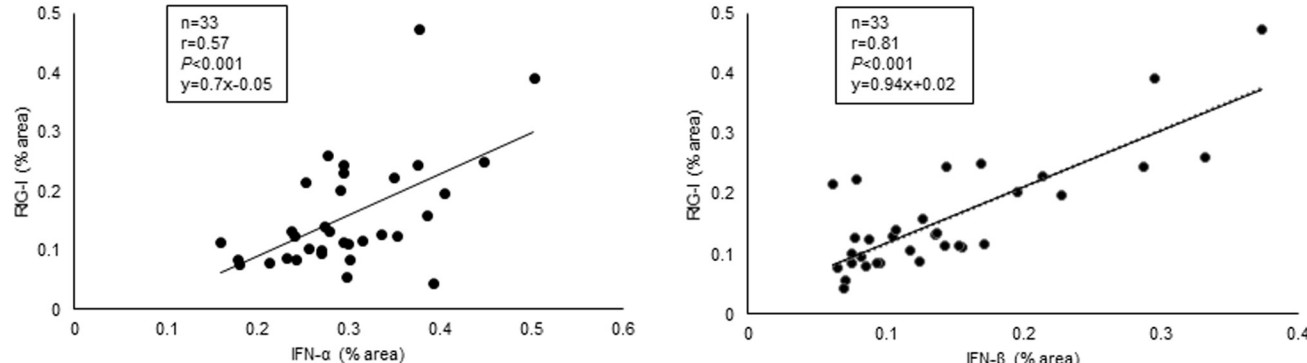

**Fig 3. RIG-I expression correlates with that of IFN-α/β in kidney biopsy specimens from IgA nephropathy patients.** Immunohistochemical staining for RIG-I and IFN-α/β was performed on 33 human kidney biopsy samples of IgA nephropathy. (A) Representative immunohistochemical staining images showing expression and localization of RIG-I and IFN-α/β in human kidney sections. (B) Expression levels of RIG-I are correlated with those of IFN-α/β (r = 0.57, $P<0.001$, and r = 0.81, $P<0.001$, respectively). Correlations were calculated using univariate correlation analysis (n = 33). Bar = 100 μm.

### IFN-α is upregulated under hypoxic conditions in Kl$^{-/-}$ mice

To identify the effects of hypoxia on the expression of IFN-α *in vivo*, we examined its expression level and localization in hypoxic WT and Kl$^{-/-}$ mice. Western blotting revealed that IFN-α expression increased in WT mice with hypoxic stimulation, and that it was further upregulated in Kl$^{-/-}$ mice (Fig 6A). Immunohistochemical staining also revealed that the IFN-α-positive area was mainly observed in tubular cells, and that the expression level of IFN-α showed the same tendency as in the western blotting (Fig 6B).

### IFN-β is increased under hypoxic conditions in Kl$^{-/-}$ mice

In addition to IFN-α, we examined the expression level and localization of IFN-β in hypoxic WT and Kl$^{-/-}$ mice. From the results of western blotting, IFN-β expression increased in WT mice with hypoxic stimulation, while a further increase was observed in Kl$^{-/-}$ mice (Fig 7A). Upon quantifying the IFN-β-positive area, the expression level of IFN-β increased in Kl$^{-/-}$ mice compared with that in WT mice (Fig 7B).

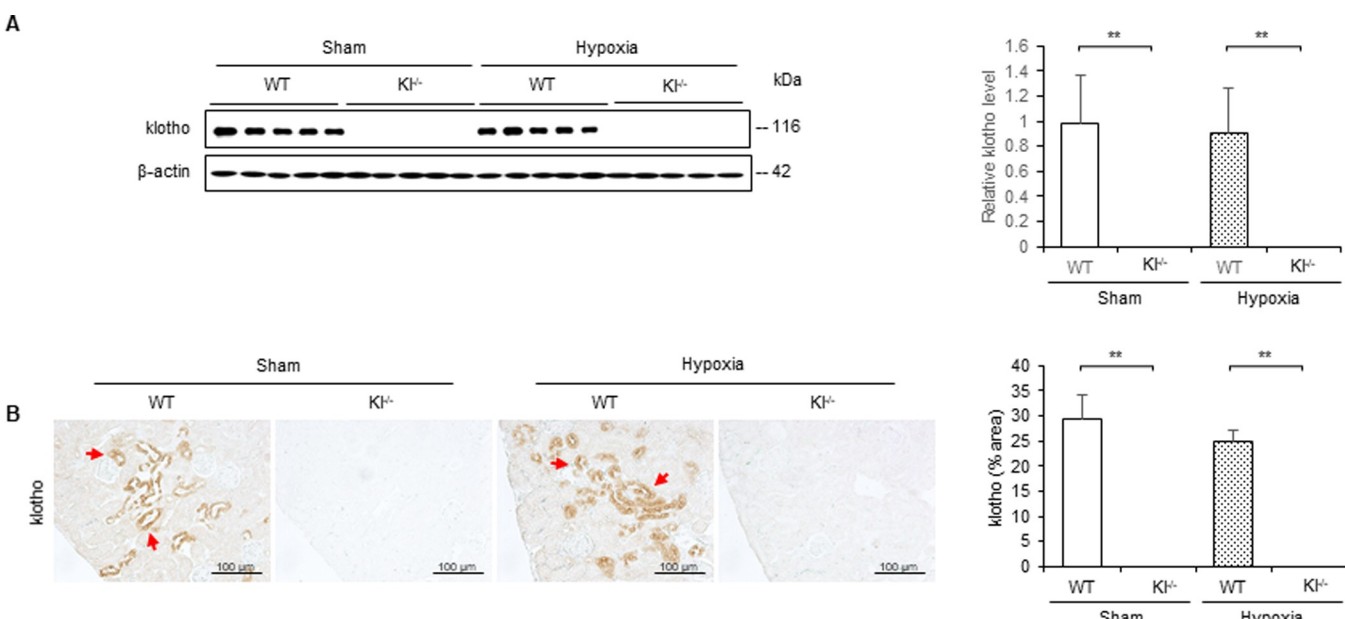

**Fig 4. Expression level of klotho does not change during 10 min of hypoxic stimulation in WT and Kl$^{-/-}$ mice.** Renal hypoxia was induced by clamping the renal artery for 10 min in WT mice and Kl$^{-/-}$ mice. (A) Western blot analysis demonstrating klotho expression in WT and Kl$^{-/-}$ mice. Protein levels were normalized to β-actin levels (n = 5 in each group). (B) Representative immunohistochemical staining images showing expression and localization of klotho in WT and Kl$^{-/-}$ mice. Values are mean ± SD. $^{*}P < 0.05$, $^{**}P < 0.01$. Bar = 100 μm.

## Discussion

In this study, we demonstrated that the expression of RIG-I and IFN-α/β was upregulated in NRK-52E cells when these cells were cultured under hypoxic conditions. We also clarified that

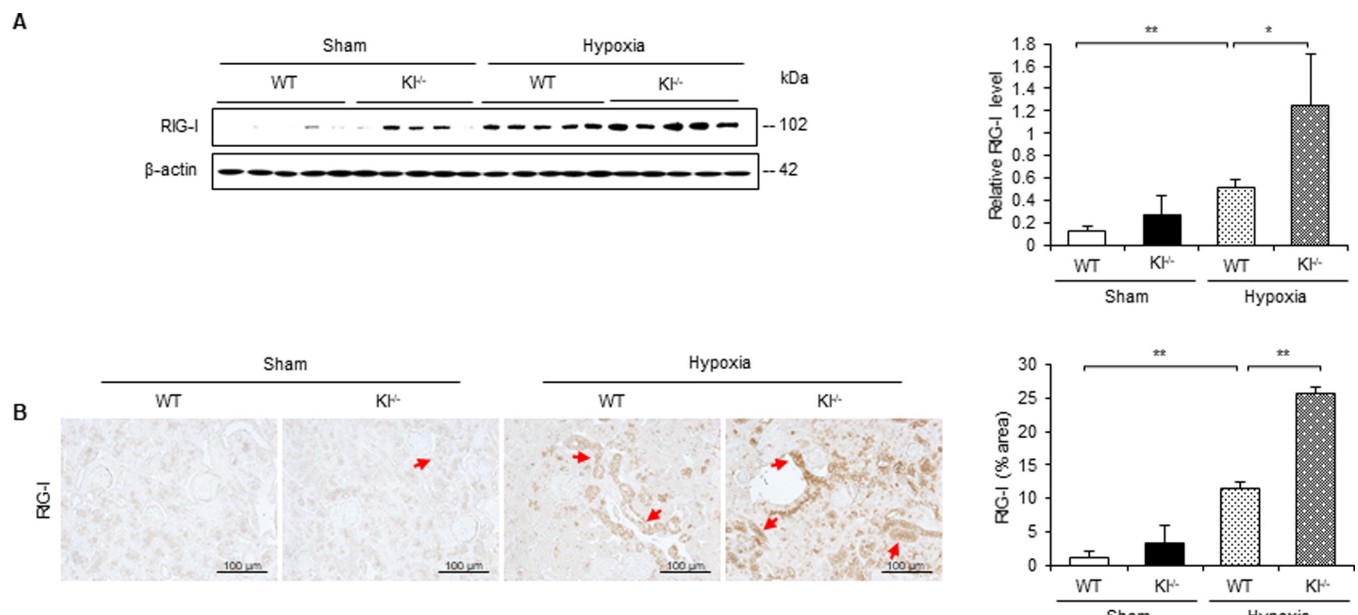

**Fig 5. RIG-I expression is intensified under hypoxic conditions in Kl$^{-/-}$ mice.** (A) Western blot analysis demonstrating RIG-I expression in WT and Kl$^{-/-}$ mice. Protein levels were normalized to β-actin levels (n = 5 in each group). (B) Representative immunohistochemical staining images showing expression and localization of RIG-I in WT and Kl$^{-/-}$ mice. Values are mean ± SD. $^{*}P < 0.05$, $^{**}P < 0.01$. Bar = 100 μm.

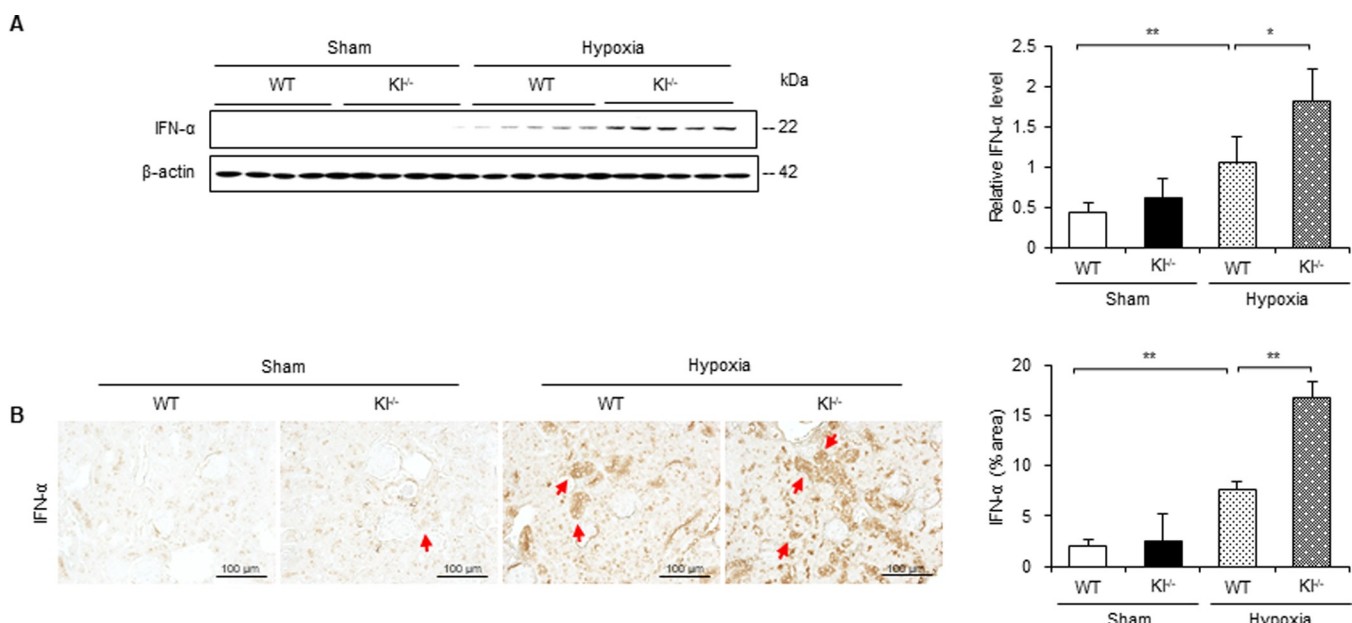

**Fig 6. IFN-α is upregulated under hypoxic conditions in Kl$^{-/-}$ mice.** (A) Western blot analysis demonstrating IFN-α expression in WT and Kl$^{-/-}$ mice. Protein levels were normalized to β-actin levels (n = 5 in each group). (B) Representative immunohistochemical staining images showing IFN-α expression in WT and Kl$^{-/-}$ mice. Values are mean ± SD. $^{*}P < 0.05$, $^{**}P < 0.01$. Bar = 100 μm.

RIG-I was responsible for the expression of IFN-α/β in *in vitro* experiments. In human biopsy samples of IgA nephropathy, the expression level of RIG-I was shown to be positively correlated with that of IFN-α/β. In *in vivo* study, we found that the renal expression of klotho did not differ between WT mice with or without hypoxic stimulation. Klotho expression was not

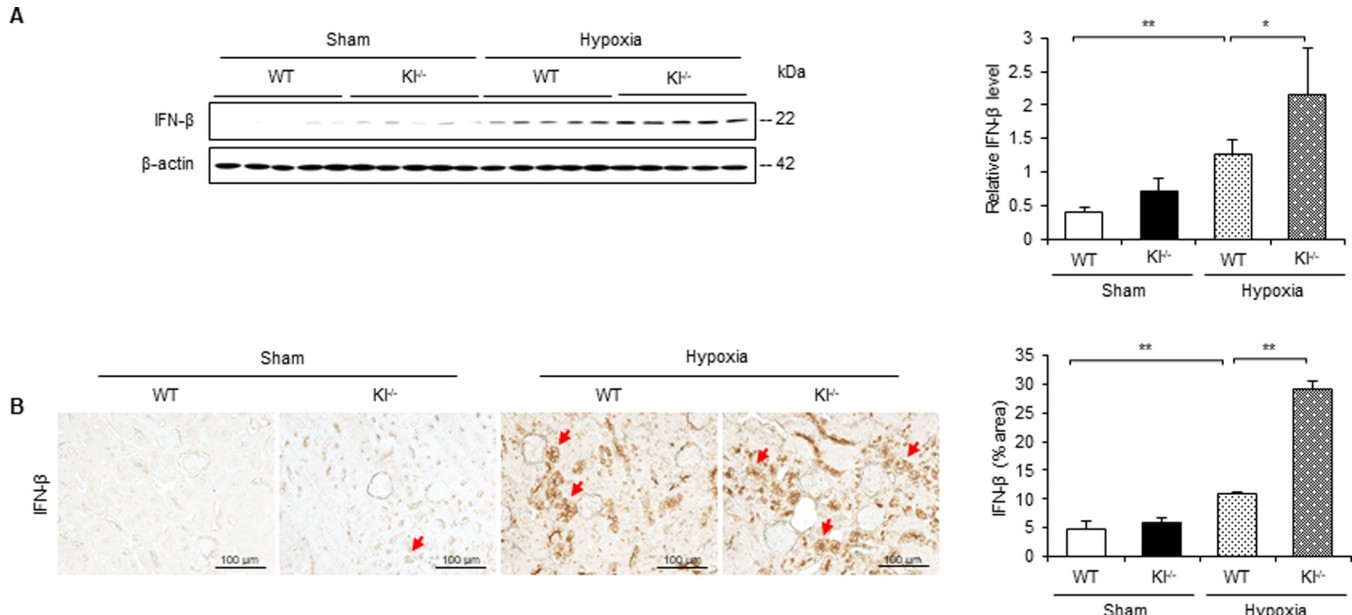

**Fig 7. IFN-β is increased under hypoxic conditions in Kl$^{-/-}$ mice.** (A) Western blot analysis demonstrating IFN-β expression in WT and Kl$^{-/-}$ mice. Protein levels were normalized to β-actin levels (n = 5 in each group). (B) Representative immunohistochemical staining images showing IFN-β expression in WT and Kl$^{-/-}$ mice. Values are mean ± SD. $^{*}P < 0.05$, $^{**}P < 0.01$. Bar = 100 μm.

observed in Kl$^{-/-}$ mice regardless of the presence of hypoxic stimulation. We showed that renal hypoxia increased the expression of RIG-I and IFN-α/β, and that their expression was further intensified in Kl$^{-/-}$ mice. These findings suggested that klotho deficiency intensified the hypoxia-induced expression of RIG-I, resulting in the upregulation of IFN-α/β.

We identified that hypoxia induced the expression of RIG-I and IFN-α/β in both *in vitro* and *in vivo* studies. To adapt to hypoxic conditions, HIF is stabilized and functions as a transcription factor [30]. In a previous study, it was reported that RIG-I is involved in HIF-inducible molecules. However, we could not identify that RIG-I is a downstream effector of HIF. In addition to virus infection, RIG-I expression increases under hypoxic conditions [31]. Although previous studies reported that hypoxia affects tubular epithelial cells, endothelial cells, pericytes, fibroblasts, inflammatory cells, and progenitor cells in kidneys [32,33], we show that the upregulation of RIG-I was mainly observed in tubular epithelial cells in mice with hypoxic stimulation, as well as in patients with IgA nephropathy. Because renal tubules are considered as an entry site of transurethral pathogens [34], RIG-I may play a pivotal role in the immune response of tubular epithelial cells under hypoxic conditions.

The innate immune response is activated by not only microorganisms but also alarm signals from damaged, injured, or stressed cells [35]. In this study, we showed that the hypoxia-induced expression of RIG-I participated in the production of IFN-α/β in a rat cell line of renal tubules. The presented data suggest that RIG-I expression functions as the innate immune system of the cytosol in response to hypoxic stress as well as virus infection. Previous studies described that hypoxia enhances during the development of CKD regardless of the primary disease, and that hypoxia contributes to the progression of renal damage [17]. In a clinical study using blood oxygenation level-dependent magnetic resonance imaging, renal hypoxia not only intensified during the progression of CKD, but also predicted such progression [36]. Thus, hypoxia is currently considered as the common pathway to the progression of end-stage kidney disease, so hypoxia-induced inflammation should be a therapeutic target to suppress CKD progression.

We showed that hypoxia induces the production of IFN-α/β, which are classified as type 1 interferons. Type 1 interferons basically function to inhibit viral replication [37] and to resist hypoxia-induced immunosuppression [21,22]. However, studies performed to date have reported that type 1 interferons lead to the activation of natural killer cells, playing an important role in not only the removal of virus-infected cells but also tissue damage [38]. In fact, previous studies reported that RIG-I contributes to inflammation in various organs, such as lung [39], kidney [40], and the nervous system [21]. In this study, we also showed that hypoxia-induced upregulation of RIG-I is responsible for the production of IFN-α/β. Thus, type 1 interferons may function to protect against infection in a state of hypoxia, but they would be harmful under uninfected conditions. Moreover, a previous study reported that, in addition to type 1 interferons, RIG-I is involved in the production of various cytokines, such as IL-1β, IL-6, and TNF-α [41]. Because these cytokines cause tissue damage, hypoxia-induced upregulation of RIG-I may participate in the progression of CKD.

We showed that klotho deficiency enhances hypoxia-induced RIG-I expression, which is accompanied by the upregulation of IFN-α/β. Klotho expression reportedly decreases with aging and the progression of CKD [42,43]. Therefore, RIG-I expression may be upregulated more in the elderly and CKD patients under hypoxic conditions, leading to the production of IFN-α/β. As mentioned above, klotho exists in three forms—membrane, secreted, and intracellular klotho—with the latter being responsible for suppressing the RIG-I-mediated production of inflammatory cytokines [26]. In terms of localization, we showed that klotho expression mainly occurs at the renal tubular cells, with evidence that RIG-I expression is mainly intensified at the renal tubular cells under hypoxic conditions. These findings suggest

that a decreased level of klotho expression contributes to enhanced RIG-I-mediated production of type 1 interferons under hypoxic conditions.

In summary, we showed that hypoxic stimulation induced RIG-I and IFN-α/β in mouse kidneys and a rat cell line of renal tubular cells, and that RIG-I was involved in hypoxia-induced upregulation of IFN-α/β in *in vitro* experiments. In renal biopsy samples from patients with IgA nephropathy, RIG-I expression correlated with the upregulation of IFN-α/β. Lastly, hypoxia-induced expression of RIG-I was intensified in klotho-knockout mice, along with the upregulation of IFN-α/β. RIG-I-mediated expression of type 1 interferon increased under hypoxic conditions, and further upregulation was observed in a state of klotho downregulation. According to clinical studies, hypoxia is exacerbated with advanced CKD stage [36] and is predictive of the long-term progression of CKD [36]. Additionally, klotho expression decreases with the decline of renal function, and reduction of klotho expression is associated with renal damage [44]. In this study, we show that klotho deficiency intensifies hypoxia-induced expression of IFN-α/β through the upregulation of RIG-I in kidneys, and that the expression of RIG-I is associated with that of IFN-α/β in actual patients with IgA nephropathy. Taking these findings together, both hypoxia and reduction of klotho are promoted during the progression of CKD, and therefore the activation of innate immunity is enhanced in CKD patients.

## Supporting information

**S1 Fig. Erythropoietin expression is intensified under hypoxic conditions in WT and Kl$^{-/-}$ mice.**
(PDF)

**S2 Fig. Hypoxia enhances expression of HIF-1α in a rat cell line of epithelial cells of renal tubules.**
(PDF)

**S3 Fig. Raw images of western blots for Fig 1A.**
(PDF)

**S4 Fig. Raw images of western blots for Fig 1B.**
(PDF)

**S5 Fig. Raw images of western blots for Fig 1C.**
(PDF)

**S6 Fig. Raw images of western blots for Fig 2A–2C.**
(PDF)

**S7 Fig. Raw images of western blots for Figs 4A, 5A, 6A and 7A.**
(PDF)

**S8 Fig. Raw images of western blots for S1 and S2 Figs.**
(PDF)

**S1 Table. Clinical characteristics related to renal function of IgA nephropathy patients.**
(PDF)

## Acknowledgments

We thank Edanz (https://jp.edanz.com/ac) for editing the English text of a draft of this manuscript.

## Author Contributions

**Conceptualization:** Asako Urabe, Shigehiro Doi.

**Formal analysis:** Ayumu Nakashima, Kensuke Sasaki, Toshiki Doi.

**Funding acquisition:** Shigehiro Doi, Takao Masaki.

**Investigation:** Asako Urabe, Takeshi Ike, Kenichi Morii.

**Methodology:** Shigehiro Doi.

**Project administration:** Shigehiro Doi, Takao Masaki.

**Resources:** Shigehiro Doi, Koji Arihiro.

**Supervision:** Shigehiro Doi, Takao Masaki.

**Writing – original draft:** Asako Urabe.

**Writing – review & editing:** Shigehiro Doi.

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
