## [Decision Letter · Decision Letter 0]

3 Jun 2021

PONE-D-21-13393

Klotho deficiency intensifies hypoxia-induced expression of IFN- α / β through upregulation of RIG-I

PLOS ONE

Dear Dr. Doi,

Thank you for submitting your manuscript to PLOS ONE. After careful consideration, we feel that it has merit but does not fully meet PLOS ONE’s publication criteria as it currently stands. Therefore, we invite you to submit a revised version of the manuscript that addresses the points raised during the review process.

The manuscript has been revised and we suggest the authors to reply point-by-point the comments raised by the referee.

We look forward to receiving your revised manuscript.

Kind regards,

Niels Olsen Saraiva Câmara, M.D, PhD

Academic Editor

PLOS ONE

Journal Requirements:

Please amend your Methods section to state the method of euthanasia/sacrifice of the rats.

Additionally, PLOS ONE now requires that submissions reporting blots or gels include original, uncropped blot/gel image data as a supplement or in a public repository. This is in addition to complying with our image preparation guidelines described at https://journals.plos.org/plosone/s/figures#loc-blot-and-gel-reporting-requirements. These requirements apply both to the main figures and to cropped blot/gel images included in Supporting Information. As such, we ask that you upload the full, uncropped images of the entire blots (not the cut strips) as a supplemental file.

PLOS ONE now requires that authors provide the original uncropped and unadjusted images underlying all blot or gel results reported in a submission’s figures or Supporting Information files. This policy and the journal’s other requirements for blot/gel reporting and figure preparation are described in detail at https://journals.plos.org/plosone/s/figures#loc-blot-and-gel-reporting-requirements and https://journals.plos.org/plosone/s/figures#loc-preparing-figures-from-image-files. When you submit your revised manuscript, please ensure that your figures adhere fully to these guidelines and provide the original underlying images for all blot or gel data reported in your submission. See the following link for instructions on providing the original image data: https://journals.plos.org/plosone/s/figures#loc-original-images-for-blots-and-gels.

4. We noticed you have some minor occurrence of overlapping text with the following previous publication(s), which needs to be addressed:

- https://ir.lib.hiroshima-u.ac.jp/files/public/4/47662/20190528114605328136/PLoSONE_13_e0202409.pdf

In your revision ensure you cite all your sources (including your own works), and quote or rephrase any duplicated text outside the methods section. Further consideration is dependent on these concerns being addressed.

Reviewers' comments:

Reviewer's Responses to Questions

**Comments to the Author**

1. Is the manuscript technically sound, and do the data support the conclusions?

Reviewer #1: Partly

Reviewer #2: Yes

2. Has the statistical analysis been performed appropriately and rigorously? 

Reviewer #1: Yes

Reviewer #2: Yes

3. Have the authors made all data underlying the findings in their manuscript fully available?

Reviewer #1: Yes

Reviewer #2: Yes

4. Is the manuscript presented in an intelligible fashion and written in standard English?

Reviewer #1: Yes

Reviewer #2: Yes

5. Review Comments to the Author

Reviewer #1: The authors performed in vitro and in vivo studies and used human biopsies to assess the effect of hypoxia in RIG-mediated IFNα/β expression in kidneys. They demonstrated that RIG-I is responsible for the hypoxia induced upregulation of IFN-α/β and that klotho deficiency intensifies this process. This is a concise, well thought out study. Although interesting, some points represent flaws, and need to be addressed.

Major

- It is not clear what the research hypothesis is. Please make it clear in the Introduction section.

- The authors should describe in Materials and Methods how they kept the NRK-52E cells under hypoxic conditions.

- Figure 1: How was the expression of RIG-I after 60, 90, and 120 min of hypoxic stimulation? Was this parameter increased in a time-dependent manner?

- Please provide evidence that cells and kidneys were under hypoxia. The authors should perform an assay with hypoxia markers (e.g. pimonidazole) or evaluate the expression of hypoxia-related genes (e.g. EPO, VEGF…).

- Hypoxia promotes activation of HIF-1a, which regulates the expression of several genes. Is there a role for HIF-1a in the regulation of renal klotho/RIG-I/IFN in this study?

- Authors should consider using a klotho inhibitor to check the expression of RIG-I and IFN in NRK-52E cells under hypoxia.

- In the Conclusion section, the Authors mentioned that “Klotho expression reportedly decreases with aging and the progression of CKD [41-42], suggesting that hypoxia-induced RIG-I upregulation likely occurs in the elderly and CKD patients”. However, the present manuscript does not provide data to support such conclusion.

Minor

- The short tittle does not match the tittle. Please correct it.

- Figure 3A: Please include the name of the groups.

Reviewer #2: The manuscript by Asako Urabe and colleagues tries to propose Klotho downregulation increases hypoxia-induced expression of IFN- α / β by overexpression of Retinoic acid inducible gene I (RIG-I). Authors showed that hypoxia induces the expression of IFN-α/β through the upregulation of RIG-I and suggesting that hypoxia-induced RIG-I upregulation likely occurs in the elderly and CKD patients.

1. In my opinion, in title and keywords would be better to add kidney.

2. Microscopic magnification as a bar or numerical value must be indicated in all Figures and Legends. It would also be helpful if the authors used arrows to point out the changes they are describing in the figure legend and text, for example the expression and localization in RIG-I in WT and Kl-/- mice.

3. Please emphasize on the clinical importance of your findings. Please discuss how your results may translate into clinical practice.

4. The author should state the dose of anesthesia.

5. In M&M, several sentences need specific references e.g. Transfection of RIG-I siRNA part. Please add

6. PLOS authors have the option to publish the peer review history of their article (what does this mean?). If published, this will include your full peer review and any attached files.

Reviewer #1: No

Reviewer #2: No

---

## [Author Response · Author response to Decision Letter 0]

28 Aug 2021

Responses to Reviewer #1

Thank you for your thorough review and constructive feedback on our manuscript. We have revised the manuscript in accordance with your comments. Our responses to each comment are shown below:

1. It is not clear what the research hypothesis is. Please make it clear in the Introduction section.

Response to comment 1

Following your suggestion, we rewrote the Introduction section.

2. The authors should describe in Materials and Methods how they kept the NRK-52E cells under hypoxic conditions.

Response to comment 2

In this study, the NRK-52E cells were seeded into 10 cm culture dishes. After growing to subconfluence, the cells were incubated in a 1% O2 Modular Incubator Chamber (MIC 101; Billups-Rothenberg, San Diego, CA) for 30, 60, 90, and 120 min. We added a description of this in the Materials and Methods section.

3. Figure 1: How was the expression of RIG-I after 60, 90, and 120 min of hypoxic stimulation? Was this parameter increased in a time-dependent manner?

Response to comment 3

The expression of RIG-I peaked at 30 min and then gradually decreased under hypoxic conditions. We replaced Figure 1A to show this.

4. Please provide evidence that cells and kidneys were under hypoxia. The authors should perform an assay with hypoxia markers (e.g. pimonidazole) or evaluate the expression of hypoxia-related genes (e.g. EPO, VEGF…).

Response to comment 4

We performed western blotting for HIF-1α in the NRK-52E cells and EPO in the hypoxic kidneys to provide evidence that the cells and kidneys were under hypoxia. HIF-1α expression increased during 120 min of hypoxic stimulation in the NRK-52E cells, while EPO expression was upregulated at 10 min of ischemia in both WT and klotho-knockout mice. We added these findings in the Supporting Information.

5. Hypoxia promotes activation of HIF-1a, which regulates the expression of several genes. Is there a role for HIF-1a in the regulation of renal klotho/RIG-I/IFN in this study?

Response to comment 5

Previous study demonstrated that RIG-I is a downstream effector of HIF-1α. However, HIF-1α expression gradually increased during 120 min of ischemia, while RIG-I expression peaked at 30 min. Moreover, siRNA for HIF-1a did not suppress the hypoxia-induced upregulation of RIG-I (data not shown). Thus, we could not identify that the upregulation of HIF-1α is responsible for RIG expression.

6. Authors should consider using a klotho inhibitor to check the expression of RIG-I and IFN in NRK-52E cells under hypoxia.

Response to comment 6

Unfortunately, klotho is not expressed in NRK-52E cells. Therefore, we could not perform an in vitro experiment using a klotho inhibitor. However, in vivo study clearly showed the effect of klotho on the expression of RIG-I and IFN-α/β under hypoxia.

7. In the Conclusion section, the Authors mentioned that “Klotho expression reportedly decreases with aging and the progression of CKD [41-42], suggesting that hypoxia-induced RIG-I upregulation likely occurs in the elderly and CKD patients”. However, the present manuscript does not provide data to support such conclusion.

Response to comment 7

We previously showed that klotho expression is associated with aging and the progression of CKD. Another study also reported this. However, to avoid confusion, we rewrote the Discussion section of the manuscript.

8. The short tittle does not match the tittle. Please correct it.

Response to comment 8

Following your suggestion, we corrected the short title.

9. Figure 3A: Please include the name of the groups.

Response to comment 9

Following your suggestion, we included the names of the groups.

Responses to Reviewer #2

Thank you for your thorough review and constructive feedback on our manuscript. We have revised the manuscript in accordance with your comments. Our responses to each comment are shown below:

1. In my opinion, in title and keywords would be better to add kidney.

Response to comment 1

Following your suggestion, we added “kidney” to the keywords and title.

2. Microscopic magnification as a bar or numerical value must be indicated in all Figures and Legends. It would also be helpful if the authors used arrows to point out the changes they are describing in the figure legend and text, for example the expression and localization in RIG-I in WT and Kl-/- mice.

Response to comment 2

Following your suggestion, we added a bar to show the microscopic magnification in all figures and legend. We also added arrows to point out the changes.

3. Please emphasize on the clinical importance of your findings. Please discuss how your results may translate into clinical practice.

Response to comment 3

Following your suggestion, we rewrote the Discussion section of the manuscript.

4. The author should state the dose of anesthesia.

Response to comment 4

We added the dose of anesthesia in the Methods section.

5. In M&M, several sentences need specific references e.g. Transfection of RIG-I siRNA part. Please add

Response to comment 5

RIG-I siRNA was used in accordance with the manufacturer’s instructions. We also added a reference on the anesthesia in the Material and Methods section.

Thank you for pointing out some issues with our manuscript. We have revised the manuscript in accordance with your comments. Our responses to each comment are shown below:

1.Please amend your Methods section to state the method of euthanasia/sacrifice of the rats. 

Response to comment 1

The method of euthanasia/sacrifice was cardiac puncture. Following your suggestion, we have added this to the Methods section.

2.We now require authors to provide the original unadjusted and uncropped images for any blot or gel data reported in PLOS ONE submissions. In our internal checks for your submission, we noted that you did not provide original raw image files supporting blot/gel data in response to our previous request.

Please provide the unadjusted and uncropped images underlying all blot and gel figures at this time; see the above URL for instructions on how the raw blot/gel image data should be prepared and submitted. 

Response to comment 2

Following your suggestion, we have provided the original unadjusted and uncropped images in the Supplemental Information.

3.We noticed you have some minor occurrence of overlapping text with the following previous publication(s), which needs to be addressed:

In your revision ensure you cite all your sources (including your own works), and quote or rephrase any duplicated text outside the methods section. Further consideration is dependent on these concerns being addressed.

Response to comment 3

As you have pointed out, some of the Methods section overlapped with our previous publication. Following your suggestion, we have rewritten the Methods section to resolve this.

---

## [Decision Letter · Decision Letter 1]

7 Oct 2021

Klotho deficiency intensifies hypoxia-induced expression of IFN- α / β through upregulation of RIG-I in kidneys

PONE-D-21-13393R1

Dear Dr. Doi,

We’re pleased to inform you that your manuscript has been judged scientifically suitable for publication and will be formally accepted for publication once it meets all outstanding technical requirements.

Kind regards,

Niels Olsen Saraiva Câmara, M.D, PhD

Academic Editor

PLOS ONE

Additional Editor Comments (optional):

Reviewers' comments:

Reviewer's Responses to Questions

**Comments to the Author**

1. If the authors have adequately addressed your comments raised in a previous round of review and you feel that this manuscript is now acceptable for publication, you may indicate that here to bypass the “Comments to the Author” section, enter your conflict of interest statement in the “Confidential to Editor” section, and submit your "Accept" recommendation.

Reviewer #1: All comments have been addressed

Reviewer #2: (No Response)

2. Is the manuscript technically sound, and do the data support the conclusions?

Reviewer #1: Yes

Reviewer #2: Yes

3. Has the statistical analysis been performed appropriately and rigorously? 

Reviewer #1: Yes

Reviewer #2: Yes

4. Have the authors made all data underlying the findings in their manuscript fully available?

Reviewer #1: Yes

Reviewer #2: Yes

5. Is the manuscript presented in an intelligible fashion and written in standard English?

Reviewer #1: Yes

Reviewer #2: Yes

6. Review Comments to the Author

Reviewer #1: (No Response)

Reviewer #2: (No Response)

7. PLOS authors have the option to publish the peer review history of their article (what does this mean?). If published, this will include your full peer review and any attached files.

Reviewer #1: No

Reviewer #2: **Yes: **Hassan Askari

---

## [Editor Report · Acceptance letter]

11 Oct 2021

PONE-D-21-13393R1 

Klotho deficiency intensifies hypoxia-induced expression of IFN-α/β through upregulation of RIG-I in kidneys 

Dear Dr. Doi:

I'm pleased to inform you that your manuscript has been deemed suitable for publication in PLOS ONE. Congratulations! Your manuscript is now with our production department. 

Kind regards, 

on behalf of

Prof. Niels Olsen Saraiva Câmara 

Academic Editor

PLOS ONE